# Financial Inclusion and Intersectionality: A Case of Business Funding in the South African Informal Sector

Munacinga Simatele [1,*] and Martin Kabange [2]

1   Economics Department, University of Fort Hare, East London 5200, South Africa
2   Postdoctoral Fellow, University of Fort Hare, East London 5200, South Africa
*   Correspondence: msimatele@ufh.ac.za

**Abstract:** Financial inclusion is a critical tool in the fight against poverty. This is especially important in economies where informal markets are prevalent due to the pervasion of market failures. Marginal identities such as gender, income and race are generally noted in the literature as factors influencing access to finance. However, these marginalities are often investigated linearly, with little attention paid to the fact that they interact to compound financial exclusion. Using a survey of informal traders, the paper investigates how having multiple marginalities influences the choice of start-up capital. A sample is drawn from three different provinces in South Africa. A multinomial logit model is estimated. Using a simulation of representative groups, the paper shows that multiple marginalities matter in accessing finance. Education emerges as the most important factor that can temper the effect of other marginalities in the financial sector. Both females and blacks with higher levels of education have better access to more stable sources of start-up capital.

**Keywords:** financial inclusion; informality; intersectionality; marginalisation

**JEL Classification:** O15; G17; G40; G41

## 1. Introduction

Informality is ubiquitous, particularly in developing countries. Research increasingly recognises that the informal sector is extensive and growing (Webb et al. 2013). The sector has proven to house vibrant businesses and provide full-time and casual wage employment and diverse activities. These activities range from light manufacturing to street vending. The International Labour Organisation (ILO 2019) shows that the informal sector contributes up to 89% of total employment in sub-Saharan Africa. Additionally, Webb et al. (2013) indicate that informal sector activities contribute 10–20% of the GDP in developed economies and up to 60% in developing economies. In South Africa, the informal economy accounts for about 17% of total employment and about 5.2% of the country's GDP (STATSSA 2021).

Despite these laudable contributions, small business owners struggle to access finance for setup and growth (Carreira and Silva 2010; Kislat 2015). It is also documented that small businesses typically use informal finance. However, very little is known about the underlying factors that contribute to choices of funding sources other than the rationing behaviour of banks (Nguyen and Canh 2020). Fraser (2009) argues that some businesses opt to use informal finance even when accessing formal finance. This paper proposes that looking at underlying factors in a linear manner can distort conclusions on the accessibility of finance. We argue that the various factors interlock to exclude even those who should ideally access finance. Furthermore, growing literature indicates that entrepreneurship is socially embedded so that social hierarchies and implied positionality are relevant to pursuing entrepreneurial opportunities (Martinez Dy 2020).

The lack of access to capital is more pronounced amongst marginalised populations such as females, minorities, the disabled and those with little or no education. The literature

on factors influencing access to finance has underlined that such entrepreneur characteristics strongly influence access to capital from various sources (Watson 2006; Sebopetji and Belete 2009; Nguyen and Luu 2013; Menike 2015). However, it would be erroneous to conceive that the influence of such factors is linear and additive. The reality is that the marginalities that individuals experience are interlocking with simultaneous influence on their access to capital. Females cannot detach themselves from their gender just because they are disabled or have no education. These various marginalities will have a multiplier effect on their inability to access capital for their businesses.

This paper applies the concept of intersectionality to investigate the effect of interlocked marginalities on access to capital amongst informal businesses in South Africa. This framing is useful, because it accounts for the effect of people's overlapping identities on explaining their experiences with social reality (Leslie McCall 2014; Hankivsky 2014). A growing body of literature investigates the entrepreneurial experiences in the context of intersecting marginalities. Examples include Martinez Dy (2020), Romero and Valdez (2016), Scott and Hussain (2019) and Valdez (2011). The theory emphasises that the simultaneous impact of multiple marginalities on experiences is larger than the sum of these effects. For instance, the literature shows that race influences access to business finance (Asiedu et al. 2012; Robb and Morelix 2016). Our results show that the effect is not linear. Instead, the effect is moderated by education, where the impact of race on access to capital is pronounced at low levels of education, but tapers off at higher levels. Similarly, we find that the level of education and age influences the effects of gender.

Therefore, a holistic approach to understanding access to entrepreneurial finance must be combined, as implied in intersectionality thinking, rather than in silos. We start the paper by reviewing the theoretical foundations for using intersectionality theory. A brief literature review on the role of entrepreneurial characteristics on access to finance, emphasising marginal social positions, follows. The data and methodology are then presented, followed by a discussion of the results and the conclusion.

## 2. Theoretical Framing

The study draws on intersectionality theory and borrows concepts from feminist literature (Leslie McCall 2014; Hankivsky 2014). Crenshaw (1989) identified three dimensions of intersectionality: structural, political and representational. These approaches address studies of social groups in society at the intersection of different categories (i.e., race and gender, location, source of funds) to explore how such groups experience marginality because of social relations and movements in which they are situated. In the context of this paper, we use the term intersectionality to show that the multiple identities encompassed within entrepreneurs influence their access to financial resources to start or expand their businesses.

Two core tenets of intersectionality underpin the framing of this study. The first tenet posits that the effects of characteristics such as age, race and gender on the individual experiences are indivisible, intersecting, mutually reinforcing and therefore non-additive (Lewis et al. 2016; Lewis and Neville 2015). The literature has investigated the effects of these characteristics in a manner that suggests additive effects. The additive analysis ignores the multiplier effects on individuals who experience more than one of these factors. Intersectionality studies, especially those in feminist scholarship, have demonstrated that race and gender are enmeshed, and their effects cannot be separated (Bailey 2009; Crenshaw 1991; McCall 2005).

The second tenet is that intersectionality framing recognises intra-group differences showing that the experiences of members of a group are not necessarily the group's experiences (Nash 2008; Crenshaw 1991). As a result, individuals can simultaneously experience both privilege and oppression. For instance, while females may have difficulty accessing finance, educated women face less of a challenge than those with less education. Similarly, women from minority or disadvantaged racial groups may be more disadvantaged (Romero and Valdez 2016; Valdez 2011). To that effect, intersectionality focuses on marginalised indi-

viduals within groups who typically come from minorities or historically disadvantaged groups. Nash (2008) argues that intersectionality forces scholars to come to terms with the legacy of exclusion. Exclusion from finance, especially credit as a source of business capital, undergirds many governments' and non-governmental organisations' attempts to increase financial inclusion as a path to growth and poverty alleviation. Accordingly, intersectionality provides a relevant theoretical framework for exploring the role of socioeconomic factors in instigating and driving financial exclusion.

The proposition to use an intersectionality lens to understand the effect of marginality on business capital stems from the understanding that markets mediate social reality. Without discarding the economic emphasis, agentic relationships between buyers and sellers primarily drive markets and depend on prices. The paper focuses on the fact that such markets only regulate how different economic agents interact with each other and do not constitute the whole. Instead, a note is taken that markets are part of larger social order. Such social order primarily leans on the definition of social relations and their associated norms of socialisation and social identities such as class, gender, race and age, and that such factors are critical for economic interaction (Etzioni 1990; Varman and Costa 2008). The intersection of these categories constrains the ease with which individuals in informal markets located on the margins of such categories or social locations can access capital.

Take social class as an example. Class implies the ownership of property and wealth. This evidences itself in the ownership of assets that can be used for collateral, bank accounts providing lending histories, and differing levels of education and their associated human capital. Valdez (2011) argues that individuals in the middle class are more likely to have access to substantial financial capital and are, therefore, better positioned than those in the lower classes to start a business. This kind of exchange defines primary exchange in capital markets. Impliedly, those in lower classes, such as those with low education and incomes, will have more constrained access to capital and are, therefore, more likely to access start-up capital through informal means.

Additional secondary relationships arising from reciprocity gathered from long-term social relationships are also important. These relationships are based on recognition and identification such as those observed in race and gender, which create conditions for the emergence of social capital (Valdez 2011; Varman and Costa 2008). Social capital can signal other market participants, thereby influencing the resources, including start-up capital, that an individual can access (Alexy et al. 2012; Packalen 2007). The influence of social capital is pronounced for microenterprises, of which most in South Africa are informal. Related, social networks tend to be segregated by race and gender. As a result, they influence the nature of social capital development and, hence, influence access to capital (Burt 1998; van Emmerik 2006).

## 3. Brief Literature Review on Informality and Sources of Funding

Various social and economic factors such as age, gender, race and education affect the ease with which individual entrepreneurs can access finance (Archer et al. 2020; Demont 2020; Nguyen and Luu 2013). Age is often negatively correlated with access to informal credit and positively correlated with access to formal finance. Older entrepreneurs are more likely to borrow from formal sources (Archer et al. 2020; Nguyen and Canh 2020). Age should affect access to finance for several reasons. First, age is likely to be positively correlated with experience. Therefore, it will introduce a positive assessment of credit worth by lenders. Second, older entrepreneurs are more likely to be risk averse. Consequently, they will seek out formal loans that are safer and more secure than informal loans.

Similarly, educational levels are negatively associated with access to informal finance. More educated owners and managers are more likely to draw up business plans, which increases the likelihood of obtaining a formal loan (Archer et al. 2020; Nguyen and Luu 2013). In addition, they are more likely to have more extensive social networks that increase the firms' social capital, which can be leveraged for increased financial capital (Nguyen and Luu 2013). Highly educated business owners are more likely to have savings and, therefore,

are more likely to be self-financing rather than resorting to informal loans (Nguyen and Luu 2013).

Gender has emerged as an important factor in access to finance. Female-headed businesses have more constrained access to start-up capital than their male counterparts (de Andrés et al. 2021; Moro et al. 2017). Despite the evidence that women have lower default rates, women are less likely to receive private financing even if they have characteristics and resources like their male counterparts (de Andrés et al. 2021; Chen et al. 2018). This is observed across different sources of capital. For example, Brush et al. (2018) show that male-managed firms are four times more likely to receive venture capital than firms managed by women in the USA. Where women can access loans, the literature suggests that they face less favourable terms than men (Alesina et al. 2013; Moro et al. 2017).

The literature attributes this to discrimination and risk aversion. First, in general, women are discriminated against in entrepreneurship, which permeates capital markets. de Andrés et al. (2021) argue that female-led start-ups are discriminated against in the loan markets. They demonstrate that female entrepreneurs are less likely to receive loans in the first year of operating despite the fact that they have better loan performance. Moreover, Muravyev et al. (2009), Qi et al. (2022) and Aristei and Gallo (2022) show that when granted loans, female entrepreneurs are charged higher interest rates than male entrepreneurs. Similarly, evidence from South Africa and India shows that women are disadvantaged in the market for small business credit (Chaudhuri et al. 2020; Zizile and Tendai 2018). This discrimination has partly been attributed to entrepreneurship's gendered nature, which takes the default position that entrepreneurship is masculine (Knight 2016; Marlow and Swail 2014). In addition, some literature, such as Blanchard et al. (2008), Moro et al. (2017) and de Andrés et al. (2021), among others, suggests that lenders do not necessarily set out to discriminate against women. Instead, discrimination arises implicitly due to the absence of relevant data. Decision makers are forced to rely on their perceptions and rationalise information regarding their profit goals, and often their perceptions are biased against women.

Women are also said to exhibit self-discriminatory behaviour. Female business owners are less likely to apply for credit because they believe they will not be successful (de Andrés et al. 2021; Ongena and Popov 2016). This self-discriminatory behaviour of women can result from a lack of confidence in their bargaining abilities, reliance on internal funding sources, or reliance on network sources such as family and friends (Alesina et al. 2013; Galli, Mascia). Some scholars argue that women are typically risk averse. As a result, their propensity to leverage businesses through external funding is lowered (Bellucci et al. 2010; Kwapisz and Hechavarría 2018).

Race has been recognised as a factor that affects access to credit. The literature shows that lenders discriminate against minority groups, and therefore these groups have higher rates of credit rationing (Asiedu et al. 2012; Bates and Robb 2016). One of the reasons for this is risk aversion and information asymmetries. To assess the risk of borrowers, lenders need information about the borrower. In the absence of information, lenders use proxies to screen. The literature suggests that one proxy used is race and its antecedents, such as neighbourhoods. The result is higher reservation prices among blacks and other ethnic minorities (Bates and Robb 2016; Faber and Friedline 2020; Fesselmeyer and Seah 2017), likely influencing the cost of loans and rationing them out of the loan market and access to start-up capital. Where there are strict regulations regarding discrimination in loan access, some scholars suggest that access can be affected by poor quality credit to markets that are predominantly minority, which affects their entrepreneurial outcomes. Additionally, access to start-up capital is also affected by entrepreneurial heritage. Michaelides (2017) indicates that whites and Asians have better access to start-up capital than blacks and Hispanics due to their family income and other related factors such as the good credit histories of their families.

Most studies ignore the interdependencies, despite the importance of these characteristics and social identities in accessing finance. In effect, this assumes that the experiences of

groups are homogeneous. However, growing literature provides evidence of the interlacing effects of social identities on entrepreneurship. For example, Romero and Valdez (2016) demonstrate that the entrepreneurial experiences of ethnic business owners differ significantly by gender, class and race. In addition, a large volume of feminist literature shows that the experiences of black female entrepreneurs differ markedly from those of black men (Wingfield and Taylor 2016). The proposal of this paper is that to effectively understand how these identities (or factors) influence access to capital, an intersectionality perspective that recognises the interlocking and independencies amongst these factors is important. The following section empirically explores this assertion using data from South Africa.

## 4. Data and Methodology

The paper adopts a categorical complexity approach suggested by McCall (2005). The approach is helpful because of its focus on the relationships of inequality amongst social groups. It allows for the disentangling of characteristics that account for heterogeneity in what are typically viewed as homogeneous groups. For example, the literature shows that women are excluded from finance without paying attention to the fact that this may not include educated or white women. Education and race are analysed as underlying factors rather than foregrounded as identities that have a significant effect. Glenn's (2002) approach of anchoring on specific categories is adopted to keep the analysis tractable. Gender, race and education are the anchor categories to capture structural differences amongst the entrepreneurs. The variables that signify marginality interact to see whether the presence of more than one marginality in an individual magnifies the effects on the ability to access capital.

The data were collected from 600 randomly selected informal businesses in three South African cities, Durban, East London and Johannesburg, in 2019. A semi-structured questionnaire was administered in a broader study investigating the impact of informality on livelihoods. This paper focuses on access to start-up capital. A total of 241 respondents did not include their source of capital. Therefore, only 359 respondents were included in the analysis. A logit model was used to investigate the probability of using a particular source of start-up capital given a set of marginalities.

Table 1 shows the demographic characteristics of the sample. The data show that 50.4% of the respondents used their savings as start-up capital. Most were females (63%) and black Africans (92.8%). The majority were involved in the selling of general merchandise. Interestingly, over 60% of them had a secondary school education, with an average of 9 years of schooling and a maximum of 16 years, in contrast to the general view that the informal sector mainly houses people with no skills and low levels of education. The average weekly income was unexpectedly high at ZAR 2639 (approximately USD 173). The amount needs to be seen in context. The bottom 50% of the sample had an average weekly income of just ZAR 622 (approximately USD 37.64), while the top 10% had ZAR 10,910 (approximately USD 660) per week. The mean age was 43 years.

**Table 1.** Data characteristics.

|  |  | **Frequency** | **Valid Percent** |
|---|---|---|---|
| **Source of Capital** | Family/friends | 102 | 28.4 |
|  | Own savings | 181 | 50.4 |
|  | Welfare/pension | 42 | 11.7 |
|  | Loans | 34 | 9.5 |
|  | Total | 359 | 100.0 |
| City | Durban | 243 | 40.7 |
|  | East London | 224 | 37.5 |
|  | Johannesburg | 130 | 21.8 |
|  | Total | 597 | 100.0 |

**Table 1.** *Cont.*

|  |  | Frequency | Valid Percent |
|---|---|---|---|
| Race[1] | Black African | 543 | 92.8 |
|  | Coloured | 1 | 0.2 |
|  | Indian | 41 | 7.0 |
|  | Total | 585 | 100.0 |
| Gender | Male | 217 | 36.6 |
|  | Female | 376 | 63.4 |
|  | Total | 593 | 100.0 |
| **Trading sector** |  |  |  |
| Valid | Fruits and Vegetables | 136 | 22.7 |
|  | General merchandise | 356 |  |
|  | Own produce/services | 108 |  |
|  | Total | 600 |  |

|  | *N* | *Minimum* | *Maximum* | *Mean* | *Std. Dev* |
|---|---|---|---|---|---|
| Age | 594 | 18 | 80 | 43.47 | 13.184 |
| Income/week | 581 | 8.0 | 70,000.0 | 2638.977 | 4938.4245 |
| Education level | 592 | 0.0 | 16 | 9.55 | 3.936 |

## 5. Interlocking Marginalities and Access to Capital

Entrepreneurs face a choice of four sources of capital $Y_j$, where $j = 1, \ldots, 4$ (family and friends, own savings, welfare/pension, loan). An entrepreneur is assumed to choose the source of capital that maximises utility subject to constraints. The constraints include market characteristics $X_i$, such as their trading sector, weekly income and location. In addition, the individual characteristics $Z_i$ that constitute the entrepreneurs' social identities, such as gender, age, race and level of education, also constrain the choice. The full list of variables used in the analysis are shown in Table 2. The likelihood that entrepreneur $i$ will choose source $j$ is estimated using a multinomial logit model.

$$Y_j = f(X_i, Z_i) \tag{1}$$

**Table 2.** Variables used in the multinomial regression.

| Variables | Description |
|---|---|
| gender | 0 = for male, and 1 = female |
| Race | 1 = black, 0 = non-black |
| Age | Number of years |
| Level of education | Number of years of schooling completed |
| Income | Total approximate weekly revenue in South African rands |
| Trading sector | 1 = fruits and vegetables 2 = general merchandise 3 = own produce/service |
| City | 1 = Johannesburg 2 = East London 3 = Durban |
| Source of capital | 1 = family/friends 2 = own savings 3 = welfare/pension 4 = loans |

If $\pi_j$ ($j = 1, 2, 3, 4$) are the likelihoods or probabilities of each one of the four sources of capital occurring, the multinomial logistic model (MLM) of the current study can be specified as:

$$ln\left[\pi_j/\pi_1\right] = \begin{array}{l} \beta_{01} + \beta_{1j}\beta_1 jGen_{1i} + \beta_{2j}Race_{2i} + \beta_{3j}Age_{3i} + \beta_{4j}Inc_{4i} + \beta_{5j}Sector_{5i} + \beta_{6j}City_{6i} \\ +\beta_{7j}Edu_{7i} + \varepsilon_{ji}i \end{array} \quad (2)$$

## 6. Presentation of Results

To check whether multiple marginalities influence the type of start-up capital vis-à-vis single marginality, we focus on gender, race, age and education. The baseline model is estimated with the marginality and the control variables (income, trading sector, and the city where the business is located). The model is then developed by including interactions of the anchor variables. Focusing on the primary effects without the interactions assumes that the effect of gender, for example, is the same for unit changes in other covariates such as education and age regardless of the values of the other variables. The intersectionality argument is that this is not the case. Therefore, the discussion focuses on the model with interaction variables shown in Table 3. Savings is used as the reference category for source of capital.

**Table 3.** Estimation results [1].

| Source of Capital | Family and Friends | Welfare | Loans |
|:---:|:---:|:---:|:---:|
| Gender [2] | 3.215 *** | 7.11 *** | 8.8 *** |
| Gender * race | 0.413 | | 0.088 * |
| Gender * school | 0.91 | 1.232 * | 1.07 |
| Gender * age | 1.0 | 0.898 | 1.079 * |
| Age | 0.961 | 1.121 *** | 0.913 ** |
| Age * race | 1.064 | 37.33 *** | 1.104 * |
| School | 1.056 | 0.794 ** | 1.068 |
| School * age | 1.004 | 1.007 * | 1.00 |
| School * race | 1.197 | 6.21 *** | 0.972 |
| Income | 1.0 * | 1.00 | 1.0 |
| Race [3] | 2.178 | 0.00 | 0.00 |
| Town [4] | | | |
| East London | 1.449 | 3.716 | 0.471 |
| Durban | 0.548 | 4.017 * | 0.694 |
| Trading sector [5] | | | |
| General merchandise | 0.767 | 0.465 * | 0.442 |
| Own produce | 0.397 ** | 0.324 * | 0.386 |
| constant | 0.338 ** | 0.032 *** | 0.092 *** |
| Number of observations: 328 | | Pseudo $R^2$ = 0.12 | |

[1] Reference group is Own Savings, [2] reference group is male, [3] reference group is black African, [4] reference group is Johannesburg, [5] reference group is Fruits and Vegetables. *** = $p$ value significant at 0.001; ** = $p$ value significant at 0.05; * = $p$ value significant at 0.1.

Adding the interaction variables introduced multicollinearity into the equation. The model was checked for multicollinearity using the variance inflation factor (VIF). The number of interaction variables gives a VIF factor of over 10. To address the problem, the continuous variables were centred in line with Irwin and McClelland (2001) and Iacobucci et al. (2017). The resulting VIF scores were small, and the centred variables were used in the analysis.

The results show that multiple marginalities influence the choice of start-up capital for informal businesses in South Africa. As a result, the level of exclusion that an individual with multifaceted disadvantages will face is likely to have a multiplier effect. For instance, being female increases the likelihood of sourcing start-up capital from welfare grants rather than savings. Relative to being male, the likelihood that females will source capital from welfare grants rather than savings increases by a factor of 1.232 for every year of schooling. The likelihood of sourcing capital from welfare grants by blacks relative to non-blacks increases by a factor of 6.21 for every year change in the level of education. Similarly, the

likelihood of sourcing capital from loans for black relative to non-black females increases by a factor of 0.088.

## 7. Predictive Margins

Predictive margins are calculated for the anchor variables to elucidate the impact of multiple marginalities on sources of start-up capital. The calculations are based on the results in Table 3 using each of the anchor variables and observing how the likelihood of selecting a given source of capital changes as the values of the related variable change. The results are then used to create profile graphs highlighting the differences in the likelihood of use. In the first instance, gender is the base entrepreneur characteristic used to create profiles of a typical male and female at different ages and levels of education. The second set of plots uses race as the anchor variable, similarly plotted against the two continuous age and education variables.

Figure 1 indicates that the interaction between gender and age most significantly affects the use of own savings and welfare grants as sources of capital. Older males are less likely to use savings and more likely to use welfare grants and pensions as a source of capital. The difference between males and females is most apparent when own savings and welfare or pensions are chosen as capital sources. The gap between males and females grows with age. For instance, the likelihood of a male using welfare as start-up capital if the business owner is 28 years old is 25%, whereas that of a female is 18.5%. However, at age 58, this increases to 95% and 36% for males and females, respectively. Similarly, at age 38, the likelihood of a male using his savings to start a business is 24%, while that of females is 51.7%. However, at age 58, the likelihood of males using savings is 4.8%, while that of females is 53%.

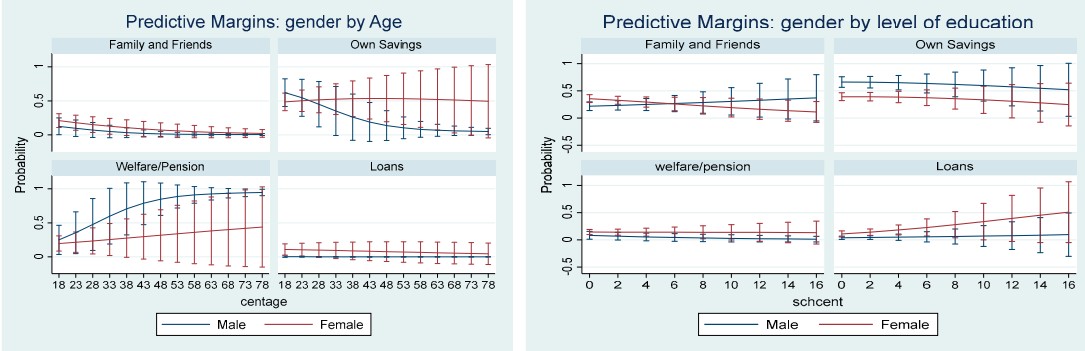

**Figure 1.** Gender interaction with education and age.

In addition, the higher the level of education, the less likely a female entrepreneur is to rely on savings and more on loans than a male entrepreneur. The use of family and friends as a source of capital dramatically decreases for females who have secondary school education (at least eight years of schooling). A female with less than six years of schooling has a 26% chance of using family and friends as a source of start-up capital. In contrast, someone who has completed secondary school with 12 years of schooling only has a 16.6% chance of sourcing capital from family and friends. The figure is lower for those with a degree, with an 11% probability of sourcing capital from family and friends.

The findings reveal that an increase in education reduces the need to rely on family and friends and on savings and welfare grants to start a business. While savings can be seen as a relatively stable source of capital, most informal business owners lack the capacity to have savings significant enough to start businesses of a meaningful size.

Figure 2 shows that older black business owners are more likely to rely on welfare and pension, while non-black businesses are more likely to rely on their savings as sources of start-up capital. The level of education makes a difference in using bank loans and own savings, although this effect is marginal in economic terms. The more educated a business

owner, the narrower the gap in their probability of using savings. Interestingly, education widens the gap between black and non-black entrepreneurs. The more educated a black entrepreneur is, the more likely it is that they will use bank loans as a source of start-up capital. For example, the probability of a black entrepreneur with six years of schooling using a bank loan as a source of start-up capital is 18%, while that of one with a degree is 36.6%. One reason could be the emphasis on historical redress in South Africa, which prioritises black business empowerment to redress the historical imbalances arising from the apartheid era.[2]

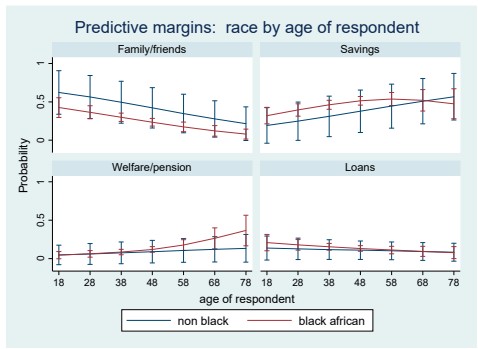 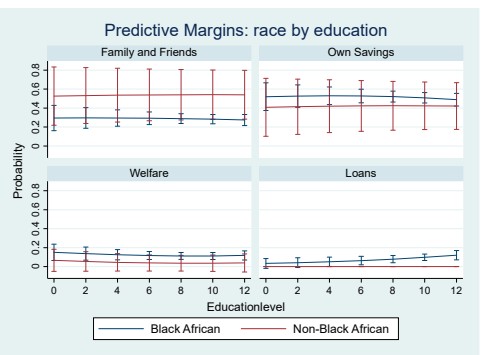

**Figure 2.** Race interaction with education and age.

## 8. Marginal Effects

The use of profile plots to visualise the interaction effects was limited to two anchor variables at a time. We developed the analysis further by calculating the marginal effects. The marginal effects are simulated for a representative entrepreneur contingent on the three anchor marginality variables as shown in Table 4.

**Table 4.** Marginal effects of a female by age and level of education.

| Age | Education Level [1] | Family and Friends | Own Savings | Welfare/Pension | Loans |
|---|---|---|---|---|---|
| 25 | No formal schooling | 0.315(0.212) | −0.366(0.165) ** | 0.082(0.155) | −0.031(0.082) |
| 25 | Completed primary school | 0.225(0.121) * | −0.312(0.109) *** | 0.104(0.055) * | −0.018(0.078) |
| 25 | Completed secondary school | 0.121(0.110) | −0.236(0.105) ** | 0.067(0.035) * | 0.048(0.077) |
| 25 | Post-secondary education | 0.017(0.111) | −0.202(0.171) | 0.040(0.033) | 0.145(0.203) |
| 45 | No formal schooling | 0.323(0.024) ** | −0.221(0.177) | −0.123(0.160) | 0.021(0.040) |
| 45 | Completed primary school | 0.187(0.082) ** | −0.318(0.086) *** | 0.068(0.046) | 0.063(0.038) |
| 45 | Completed secondary school | 0.078(0.069) | −0.315(0.074) *** | 0.110(0.035) *** | 0.128(0.037) *** |
| 45 | Post-secondary education | −0.018(0.111) | −0.303(0.124) *** | 0.113(0.047) *** | 0.208(0.098) ** |
| 65 | No formal schooling | 0.156(0.066) *** | 0.198(0.153) | −0.397(0.213) * | 0.044(0.051) |
| 65 | Completed primary school | 0.136(0.068) ** | −0.023(0.145) | −0.197(0.153) | 0.084(0.041) ** |
| 65 | Completed secondary school | 0.053(0.095) | −0.180((0.166) | 0.004(0.171) | 0.123(0.066) * |
| 65 | Post-secondary education | −0.038(0.129) | −0.260(0.201) | 0.139(0.206) | 0.158(0.132) |
| | Number of Observations: 328 | | | Pseudo R[3] 0.12 | |

[1] Representative values of education: no formal education: education level = 0, completed primary school: education level = 7 years, completed secondary school: education level = 12 years. *** = $p$ value significant at 0.001; ** = $p$ value significant at 0.05; * = $p$ value significant at 0.1.

The table shows the marginal effect of a female at representative ages and levels of education. Therefore, the marginal effects show the change in the probability of choosing the said source of capital for the female relative to the male. Four results are noted. First, the marginal effect varies at different levels of education and age for each of the four sources of capital. Second, family and friends as a source of capital are important for entrepreneurs with low levels of education. The marginal effects are only significant for those with no formal education and, at most, a primary school education. For example, a young female with a primary school education is 22.5% more likely to source their start-up capital from family and friends than a young male with the same characteristics. Similarly,

a middle-aged entrepreneur with no formal education is 32.3% more likely to source capital from family and friends.

Third, higher education levels are important in choosing to use loans as a source of start-up capital, but only for middle-aged and older entrepreneurs. Females with at least a secondary school education are more likely to use bank loans as start-up capital than males with requisite characteristics. Fourth, females are less likely to use savings as a source of capital relative to males with comparable characteristics.

These findings are consistent with the literature. For instance, Marchese and Potter (2014) show that post-secondary education minimises barriers to women's access to start-up capital. In addition, education significantly influences women's perceptions of entrepreneurship and their access to start-up capital (Halabisky and Potter 2017). Moreover, there is evidence that gender and education interact to modify access to business finance (Sena et al. 2012). Similarly, the literature on risk-taking shows that education affects the impact of gender on access to finance (Bannier and Neubert 2016).

Marginal effects were also calculated to explore how race interacts with the anchor variables of gender and education. Table 5 shows no significant difference in the probability of sourcing capital from family and friends and own savings between blacks and non-blacks. The variable is significant for welfare grants and loans. Non-blacks are less likely to use welfare and loans as a source of capital regardless of gender. However, this difference is lower for females at all levels of education. For instance, a non-black male with no formal education is 28.1% less likely to source capital from a welfare grant than an equivalent black male, while the marginal effect for a non-black female with no education is only 12.9% compared to a black female with similar characteristics. Non-blacks are also less likely to use loans. The likelihood is higher for females at all levels of education. Some scholars have noted the effect of the intersection between race and gender in accessing business finance. For instance, Sena et al. (2012) and Kwapisz and Hechavarría (2018) show that non-white women tend to self-exclude from external business finance. Bewaji et al. (2015) also show that education moderates ethnic minorities' access to start-up capital.

**Table 5.** Representative non-black entrepreneur by education.

| Highest Level of Education [1] | Gender | Family and Friends | Own Savings | Welfare | Loan |
|---|---|---|---|---|---|
| No formal education | Male | −0.075(0.217) | 0.375(0.246) | −0.281(0.136) ** | −0.019(0.035) |
| Completed primary school | Male | 0.060(0.273) | 0.095(0.275) | −0.122(0.045) *** | −0.034(0.024) |
| Completed secondary school | Male | 0.274(0.175) | −0.155(0.232) | −0.072(0.040) * | −0.047(0.040) ** |
| Post-secondary education | Male | 0.436(0.303) | −0.328(0.307) | −0.048(0.040) | −0.060(0.054) |
| No formal education | Female | −0.273(0.492) | 0.431(0.495) | −0.129(0.077) * | −0.030(0.026) |
| Completed primary school | Female | −0.041(0.302) | 0.231(0.301) | −0.109(0.030) *** | −0.081(0.034) ** |
| Completed secondary school | Female | 0.154(0.232) | 0.080(0.233) | −0.076(0.032) ** | −0.158(0.033) *** |
| Post-secondary education | Female | 0.309(0.383) | −0.005(0.386) | −0.052(0.055) | −0.251(0.092) *** |
| *Number of observations: 328* | | | | *Pseudo $R^2$: 0.12* | |

[1] Representative values of education: no formal education: education level = 0, completed primary school: education level = 7 years, completed secondary school: education level = 12 years. *** = $p$ value significant at 0.001; ** = $p$ value significant at 0.05; * = $p$ value significant at 0.1.

## 9. Conclusions and Implications

The results confirm the importance of gender, race and education as important factors that affect access to capital. More importantly, the results show that multiple marginalities matter in accessing finance. Education emerges as an important variable that can temper the effect of other marginalities in the financial sector. While the results confirm that females are more vulnerable in that they are more likely to use unstable sources of capital, females with higher levels of education are less vulnerable. Similarly, education influences the effect of race on access to bank loans. The impact of age is also important in that older black entrepreneurs are more likely to use less stable sources of capital. This is likely to be a factor unique to South Africa. Historically, blacks in South Africa had fewer opportunities for

education and were only allowed the opportunity to obtain specific types of jobs. For that reason, their skill sets are less likely to be developed, limiting their access to more low-cost sources of finance. The result on age could also be explained by the historical exclusion from education. The impact of education is significant; more educated black entrepreneurs have a greater opportunity to obtain low-cost sources of finance.

The main implication of the results is that access to start-up capital can be improved by considering the multiple social positions that entrepreneurs find themselves in. The literature shows that programmes that target specific groups including women and ethnic minorities, such as giving them direct access to finance have limited success. We suggest that these efforts could be more successful if they are complemented by improved access to education for women and other minorities. Given the possibility that women self-exclude from external financing, policy efforts should also develop programmes to engender confidence among female entrepreneurs.

An important factor in determining the source of capital is the size of capital required. These data were not captured for most of the respondents. Therefore, the paper's results need to be seen with this caveat in mind. Nevertheless, given the mean income, we believe including income size would not fundamentally change the conclusions of the analysis. The results in this paper underline the importance of using intersectionality as an analytical research lens for access to capital. Very little research exists that takes this approach in entrepreneurial finance. One key impediment highlighted in the literature is the difficulty of operationalising the intersectionality approach in empirical research. Developing methods and measurement, especially in informal markets, would ease investigation. The use of numerical data, as is done in this study, is helpful. However, the need to capture other aspects of marginalisation, such as class, may need more accurate representation, perhaps through the development of relevant scales or the incorporation of some qualitative analysis through mixed-methods approaches.

**Author Contributions:** Conceptualization, M.S.; Formal analysis, M.S. and M.K.; Investigation, M.S.; Project administration, M.S.; Writing–original draft, M.S. All authors have read and agreed to the published version of the manuscript.

**Funding:** Research Niche Area, Management and Commerce, University of Fort Hare.

**Institutional Review Board Statement:** The study was conducted in accordance with the Declaration of Helsinki, and approved by the University of Fort Hare Research Ethics Committee (SIM0001, 5 March 2018).

**Informed Consent Statement:** Informed consent was obtained from all subjects involved in the study.

**Data Availability Statement:** Not applicable.

**Conflicts of Interest:** The authors declare no conflict of interest.

## Notes

1.   There is very little variation in race. However, the model statistics indicate that race should be included.
2.   The South African government enacted the Broad-Based Black Economic Empowerment Act (2003) to advance economic transformation and participation of black people in the South African economy. The financial sector complies with the Act, resulting in this advantage for black entrepreneurs in the loans market as education increases.
3.   The South African government enacted the Broad-Based Black Economic Empowerment Act (2003) to advance economic transformation and participation of black people in the South African economy. The financial sector complies with the Act, resulting in this advantage for black entrepreneurs in the loans market as education increases.

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
