# Peer review of "Financial Inclusion and Intersectionality: A Case of Business Funding in the South African Informal Sector"

_jrfm, doi:10.3390/jrfm15090380_

Round 1

Reviewer 1 Report

The chosen topic is very interesting and topical considering the increasingly intense concerns of financial institutions regarding the development of digital payments. The approach is very interesting from the perspective of the marginalities that have an impact on financial inclusion. The article is well written, the ideas are presented in a logical order and are well supported by numerous recent references (appropriate for the chosen topic) from various sources. The manuscript has a very good potential to be published, but the authors need to make some improvements

1.       In the abstract, the authors must specify for what period the analysis was performed

2.       In the introduction, the authors could present (1) the impact of remittances on financial inclusion and (2) money laundering - financial inclusion relationship.

3.       In the data and methodology section , the authors must specify for what period the analysis was performed

4.       In the discussion section, the authors should present the results of similar studies which confirm to refute the results obtained.

5.        In the onclusions section, some policy proposals, research limits and future research directions should be inserted.

Good luck to the authors in completing this special scientific endeavor.

Author Response

Thank you for the time spent reading my manuscript and for the comments that you have provided. I have done my best to respond to all of them. a point by point response is contained in the attached letter.

Reviewer 2 Report

Major comments

1.      Your premise relates to the joint probability of two or more marginalized groups.

2.     Your hypothesis is unclear.

3.     Page 4 – is age also negatively associated with informal finance?

4.     How does the size of the capital or loan play into the analysis?

5.     Table in page 8 – I don’t know what family and friends, welfare, and loans refer to.

6.     Also, only use marginal effects (change in Y/change in X).  Otherwise, I don’t know how to evaluate the coefficients in table 3

7.     Note sure how the predictive margins were generated for the figures.

8.     Marginal effects table should include standard errors, as well as rows for number of observations and r-squared.

9.     Use the standard notation in table 4 of stars versus bold for significance

10.  Tables 4 and 5 are confusing and I don’t believe are done correctly.  It looks like you have converted a continuous variable of years of education into a categorical variable (no school, primary, secondary, post secondary).  If you use a categorical variable, you need to drop one of them and interpret the results relative to the dropped variable.

It would be better to simply take the marginal effects overall (1st regression result table) and then restrict the sample by age (second regression result table) if that is what you want to do.

11.  It is also unclear to me what you have done with age.  You show results for ages 25, 45, and 65.  However, you have many more years, so I don’t know what you did, and you don’t list observations, so I don’t know how you segmented your sample.

12.  Change the word vulnerable in your conclusions.  If females rely on family and friends more than males, it does not make them more vulnerable.

Minor comments

1.       Abstract –change multiple marginalities to multiple groups of marginalized individuals.  Also, change marginalities to marginal groups

2.      Introduction – reduce number of stats, perhaps by saying the “ informal sector activities contribute 10–20% of the GDP in developed economies and up to 60% in developing economy.”  Then delete the rest of the paragraph.

3.      Take out the word laughable

4.      Change entrepreneur to “entrepreneurial” on page 3 in the last paragraph of the introduction

5.      Page 5 – change literature suggests to scholars or authors suggest

6.      Page 5 – rewrite, “This self-discriminatory behaviour can result from a lack of confidence in own bargaining abilities, reliance on internal funding sources, or reliance on network sources such as family and friends.”  Instead say, “This self-discriminatory behavior of women can result from a lack of confidence in their bargaining abilities or from reliance on internal funding sources, or reliance on network sources such as family and friends.”

7.      Page 6 – suggest by McCall (2005).

8.      Page 7 and 8, change probabilities to likelihood.  Likelihood is the proper term to use with a logit model.

9.      Use 0,1 for black or not

10.  Page 8, change odds to likelihood.  Odds and likelihood are not the same thing.  You need to use likelihood with a logit model

11.  Also, don’t reference that the likelihood differs, but tell the reader in which direction it differs.

Author Response

(The authors gave the same response as above.)

Round 2

Reviewer 2 Report

See comments

Author Response

Thank you for reviewing the paper further. we have made the requested changes and attached an itemised response.
